# DIVERSITY REGULARIZATION IN DEEP ENSEMBLES

**Changjian Shui, Azadeh Sadat Mozafari, Jonathan Marek, Ihsen Hedhli, Christian Gagné**
Computer Vision and Systems Laboratory / REPARTI and Big Data Research Centre
Electrical and Computer Engineering Department, Université Laval, Québec (Québec), Canada
`{firstname.lastname.1}@ulaval.ca`, `christian.gagne@gel.ulaval.ca`

## ABSTRACT

Calibrating the confidence of supervised learning models is important for a variety of contexts where the certainty over predictions should be reliable. However, it as been reported that deep neural network models are often too poorly calibrated for achieving complex tasks requiring reliable uncertainty estimates in their prediction. In this work, we are proposing a strategy for training deep ensembles with a diversity function regularization, which improves the calibration property while maintaining a similar prediction accuracy.

## 1 INTRODUCTION

Probability calibration is important role in many applications of machine learning, as a reliable confidence estimation over the predictions that are made is often needed. For example, in end-to-end learning for self-driving vehicles (Bojarski et al., 2016; Richter & Roy, 2017), an accurate confidence over the detected objects is required by the neural network to predict the driving directions. Besides, *fairness* in machine learning, meaning no bias and discrimination in predictions, is closely related to the calibration, which causes concerns recently (Pleiss et al., 2017).

In the general framework of neural network, the output of *softmax* function at the last layer is typically used as confidence probability. However, it has been shown by Guo et al. (2017) that modern deep neural networks are poorly calibrated when learning complex tasks. Moreover, Lakshminarayanan et al. (2017) reported that an ensemble of deep neural networks averaging the softmax output can improve calibration. Nevertheless, from empirical results reported in the following sections, we find that ensembles are *under confident*, with confidence scores proportionally lower than their corresponding prediction accuracy. We also observe that calibration gets worse when we increase the number of members in the ensemble. Therefore, finding a calibration strategy is essential to render proper confidence estimation with deep networks ensembles.

An important aspect of ensemble learning is the *diversity* among the members. Liu & Yao (1999) proposed an explicit approach to enforce diversity between members of neural networks ensembles, by using *Negative Correlation* (NC) as a regularization component of the loss function. However, this proposal was analyzed in the perspective of improving generalization of neural networks ensemble, ignoring the calibration properties. In our work, we show that by enforcing diversity among ensemble members using NC regularization, we can improve significantly calibration property compared to pure ensemble approaches, with little or no impact on accuracy.

## 2 EVALUATING CALIBRATION

The deep neural network produces classification decision in a probabilistic form with $\hat{P}(y_t|\mathbf{x}_t; \theta)$, where $\theta$ are the weights in the neural network learned on the training set. An algorithm with perfect calibration is defined as $P(y_t = k|x_t; \hat{P}(y_t = k|\mathbf{x}_t; \theta) = p) = p$, which means that the prediction confidence for label $k$ should be equal to the accuracy on the same label. In practice, empirical approaches can be used for estimating the calibration, foremostly *reliability diagram* and *expected calibration error*.

**Reliability diagram** (DeGroot & Fienberg, 1983; Guo et al., 2017) is an accuracy-confidence function that is approximated by estimating the accuracy and confidence in $Q$ equal interval bins

---

**Algorithm 1** Training of an Ensemble of Deep Networks with Diversity Regularization

---

**Input:** Training set $\mathcal{X} = \{(\mathbf{x}_1, y_1), \ldots, (\mathbf{x}_N, y_N)\}$, $M$ neural networks $\mathrm{h}_{1:M}$
**for** each minibatch $\mathcal{X}_l \subset \mathcal{X}$ **do**
    Train each network $\mathrm{h}_i$ on minibatch $\mathcal{X}_l$ by minimizing the loss function:
    $E_i((\mathbf{x}_j, y_j); \mathrm{h}_i) = L(y_j, \mathrm{h}_i(\mathbf{x}_j)) + \lambda \operatorname{div}(\mathrm{h}_i(\mathbf{x}_j); \mathrm{h}_{1:M}), \forall (\mathbf{x}_j, y_j) \in \mathcal{X}_l$
**end for**
Prediction for the new data $x$: $\bar{\mathrm{h}}(\mathbf{x}) = \frac{1}{M} \sum_{i=1}^{M} \mathrm{h}_i(\mathbf{x})$.

---

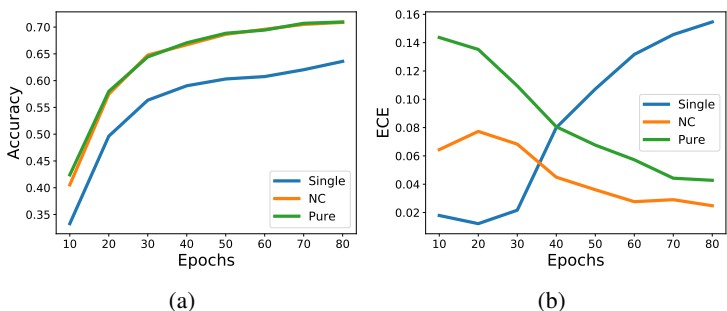

(a)            (b)

Figure 1: (a) Test accuracy and (b) Expected Calibration Errors (ECE) for different strategies during the training. (Single: one neural network; Pure: deep ensemble with independent neural networks; NC: deep ensemble with NC regularization).

$C_i = \{\hat{y}_j \mid \hat{y}_j \in [c_{i-1}, c_i), \forall \hat{y}_j\}$, uniformly covering domain $[0, 1]$, with accuracy and confidence defined as: $\operatorname{acc}(C_i) = \frac{1}{|C_i|} \sum_{\hat{y}_j \in C_i} \mathbf{1}[y_j \in [c_{i-1}, c_i)]$, $\operatorname{con}(C_i) = \frac{1}{|C_i|} \sum_{\hat{y}_j \in C_i} \hat{P}(\hat{y}_j | \mathbf{x}_t; \theta)$.

**Expected Calibration Error** (ECE) (Naeini et al., 2015) defines a metric for evaluating the calibration quality corresponding to the empirical expectation of $|\operatorname{acc} - \operatorname{con}|$, that is: $\operatorname{ECE} = \sum_{i=1}^{Q} \frac{|C_i|}{Q} |\operatorname{acc}(C_i) - \operatorname{con}(C_i)|$.

## 3 METHODOLOGY

We propose a diversity regularization training strategy, presented in Algorithm 1, where $L(y, \mathrm{h}_i(\mathbf{x}))$ is the classification error (we adopt the cross-entropy loss in our work) and $\operatorname{div}$ is the diversity metric of the ensembles, that is negative correlation:

$$\operatorname{div}(\mathrm{h}_i(\mathbf{x}); \mathrm{h}_{1:M}) = (\mathrm{h}_i(\mathbf{x}) - \bar{\mathrm{h}}(\mathbf{x}))[\sum_{j \neq i}(\mathrm{h}_j(\mathbf{x}) - \bar{\mathrm{h}}(\mathbf{x}))]$$

The $\bar{\mathrm{h}}(\mathbf{x}) = \frac{1}{M} \sum_{i=1}^{M} \mathrm{h}_i(\mathbf{x})$ is regarded as a constant with respect to $\mathrm{h}_i$ during the backpropagation.

## 4 RESULTS

**Datasets and settings** We evaluate our strategy on CIFAR-100 dataset. We train the VGG network with 11 layers and batch normalization. We initially set member size $M = 7$ and regularization parameter $\lambda = 0.1$.

**Ensemble with NC vs pure ensemble** Results of accuracy and confidence are presented in Fig. 1 and 2. We find that deep ensembles improve the accuracy and calibration compared to a single deep network. Ensemble with NC regularization reduces the ECE without losing the accuracy during the whole training procedure, as compared with the pure ensemble. Fig. 2 also presents prediction distribution and reliability diagram.

In order to directly compare the results, we pick up the *people* superclass of CIFAR-100, which includes five classes: baby, boy, girl, man, and woman. We compute accuracy and average confi-

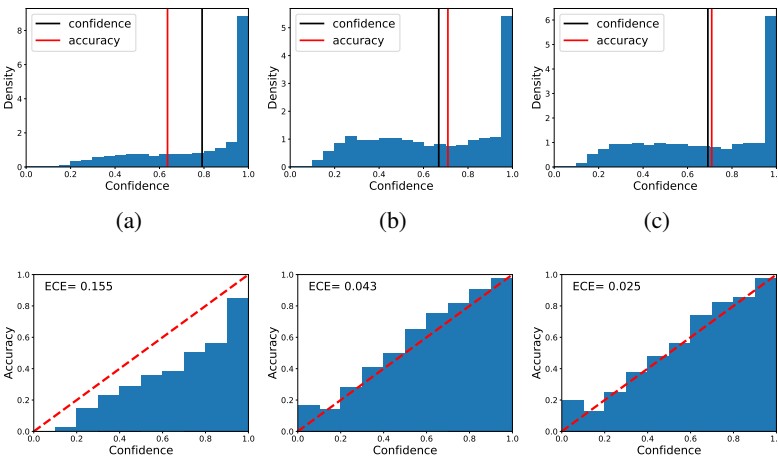

Figure 2: The distribution of predictions and reliability diagram for (a) Single deep NN, (b) Pure deep ensembles, (c) Deep ensembles with NC regularization

| | BABY | BOY | GIRL | MAN | WOMAN | AVERAGE |
|---|---|---|---|---|---|---|
| SINGLE | 0.71 (0.88) | 0.68 (0.71) | 0.81 (0.85) | 0.42 (0.64) | 0.54 (0.66) | 0.12 |
| PURE | 0.79 (0.78) | 0.76 (0.66) | 0.84 (0.76) | 0.44 (0.47) | 0.51 (0.54) | 0.05 |
| NC | 0.79 (0.78) | 0.68 (0.68) | 0.83 (0.78) | 0.51 (0.52) | 0.57 (0.59) | **0.02** |

Table 1: Accuracy and prediction confidence (inside parentheses) for superclass *people*. "AVERAGE" is the average difference between the confidence and accuracy. The values in blue and green means a significantly over confident and under confident response, respectively.

dence for each class, as shown in Table 1. We find that the prediction in single VGG is always *over confident* – the prediction confidence is generally higher than the accuracy. Simultaneously, the pure ensemble is significantly under confident for the classes boy and girl. Nevertheless, training with NC-based regularization maintains good prediction accuracy and control calibration at good level.

**Ensemble size** We tested different ensemble size $M$ and compute their corresponding test accuracies and ECE, as shown in Table 2. We find that the performance of pure ensembles is slightly better for very small $M$ on CIFAR-100. While deep ensemble with NC can effectively reduce ECE for relatively large $M$. See the appendix for detailed results with different member size.

**Other dataset** We repeat the experiments with CIFAR-10 using a modified *Alexnet*. We find that our approach can still improve the calibration and accuracy, but that a single network does not suffer from a serious calibration problem. The possible reason is the different training size for each class and number of classes. For a detailed comparison see the appendix.

**Conclusion and outlook** We apply Negative Correlation (NC) as a regularization function for training deep ensembles. Compared to a single neural network and pure deep ensembles, ensembles with NC regularization showed a better calibration property with maintaining a good accuracy. As future work, we suggest analyzing the calibration problem in few shot learning setting and transfer learning with the real images, where each class is represented with small number of instances in the training set.

| | $M = 3$ | $M = 7$ | $M = 11$ |
|---|---|---|---|
| PURE | **0.6847 (2.3%)** | 0.7088 (4.3%) | 0.7146 (6.9%) |
| NC | 0.6832 (3.5%) | **0.7096 (2.5%)** | **0.7153 (3.9%)** |

Table 2: Accuracy and ECE (values inside parentheses) for different member sizes

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

APPENDIX

### RESULTS ON CIFAR-10

We applied a modified *Alexnet* that adds batch normalization and reduces the hidden units in fully connected layers ($256 \rightarrow 128 \rightarrow 128 \rightarrow 10$). The single neural network exhibits a much better prediction calibration, whereas the pure ensemble suffers from a strong *under-confident* issue. Our proposed approach can maintain a good accuracy similarly as the pure ensemble, with an advantage of better calibration.

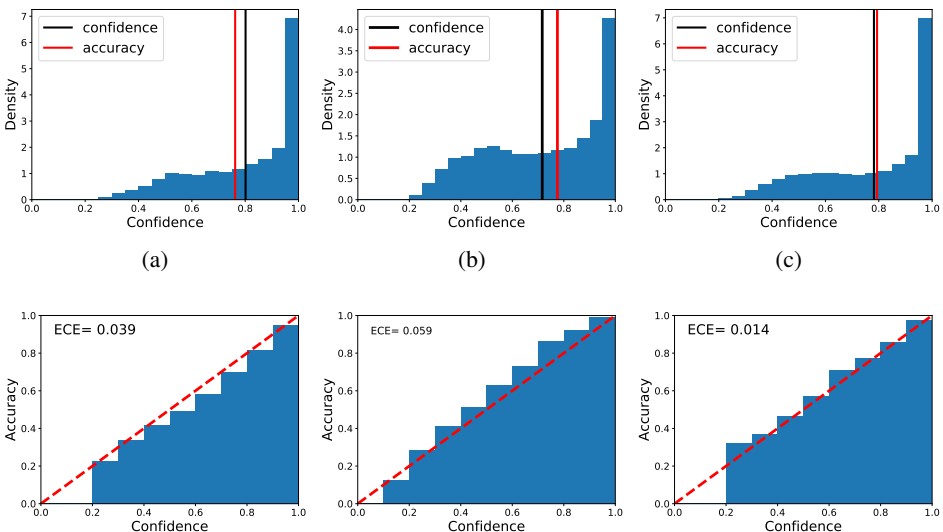

Figure 3: The distribution of predictions and reliability diagram in (a) Single CNN, (b) Pure deep ensembles $M = 3$, (c) Deep ensembles with NC regularization with $M = 3$

| APPROACH | ACCURACY | ECE |
|---|---|---|
| SINGLE | 0.7607 | 3.9% |
| PURE | 0.7751 | 5.9% |
| NC | **0.7865** | **1.4%** |

Table 3: Accuracy and ECE in different training strategies

### DEEP ENSEMBLES WITH DIFFERENT $M$

Apart from $M = 7$, we tested a very small ensemble $M = 3$ and a relatively large ensemble $M = 11$. For $M = 3$, the NC regularization does not show the regularization influence because of the small $M$. For a larger $M$, we find that the effect of NC is more obvious in reducing the *under confident* in pure ensembles.

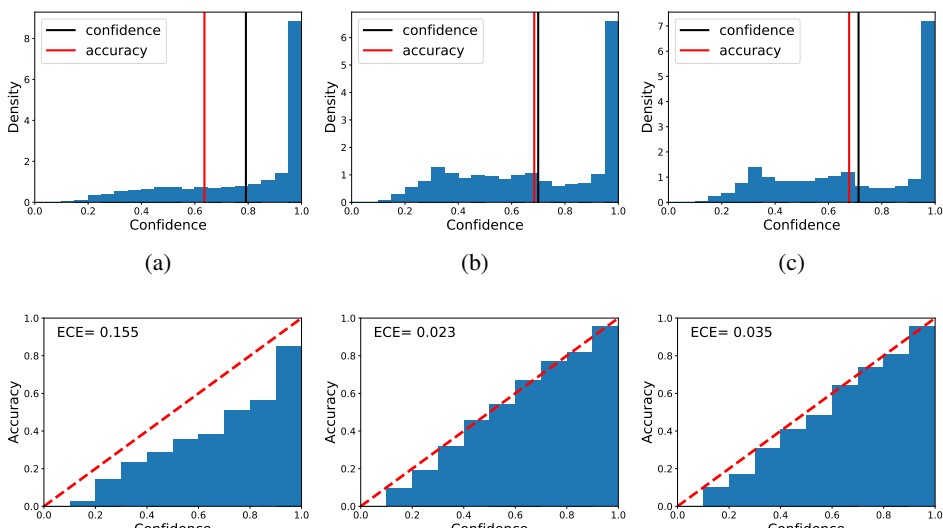

Figure 4: The distribution of predictions and reliability diagram in (a) Single CNN, (b) Pure deep ensembles $M = 3$, (c) Deep ensembles with NC regularization with $M = 3$

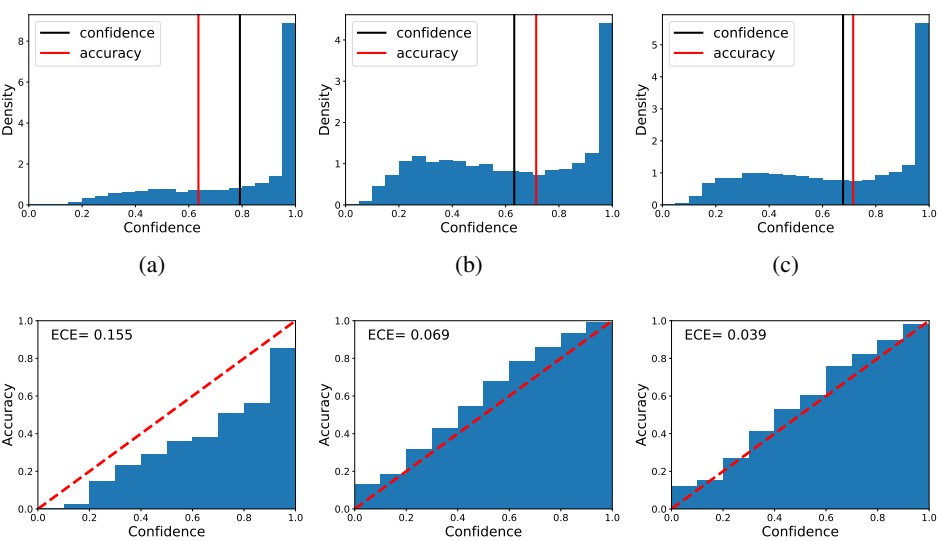

Figure 5: The distribution of predictions and reliability diagram in (a) Single CNN, (b) Pure deep ensembles $M = 11$, (c) Deep ensembles with NC regularization with $M = 11$

