# OpenReview forum: "Diversity Regularization in Deep Ensembles"
_ICLR.cc/2018/Workshop — Reject_

### Official Review · AnonReviewer2 · 2018-03-06
**A good but limited proof of work**

**Rating:** 5
**Confidence:** 3

**Review:**

Summary:
This work applies negative correlation as a regularization term for training
deep ensembles. Results show better calibration of the output probability values
compared to a single neural network and to pure deep ensembles, while maintaining
good accuracy (at least on shown results).

Assessment:
- Novelty: limited, as the method is identical to Liu and Yao (1999), but re-examined in the context of probability calibration.
- Clarity: the paper is easy to understand.
- Significance: this work provides a solution to an important problem.
- Quality: results are shown, but they are not compared to any calibration method besides ensembling.

Decision:
+ A simple idea to an important problem.
- The method is not novel, although its application to probability calibration is.
- The method is not compared to any of the many well-known methods for probability calibration (e.g, histogram-based calibration, sigmoid calibration, isotonic regression, etc). It is therefore difficult to assess the significance and quality of this work.
- Experiments are limited to a single dataset.

Overall, this work is a good proof-of-work but the lack of experimental results (on more than one dataset, and compared to well-established baselines) makes it difficult to determine the actual significance of the contribution. For these reasons, I do not recommend this paper for acceptance.

---

### Official Review · AnonReviewer3 · 2018-03-10
**A rather simplistic method for uncertainty calibration of deep networks**

**Rating:** 5
**Confidence:** 3

**Review:**

This work studies the problem of uncertainty calibration and proposes to use diversity regularization for generating deep ensembles for the purpose of uncertainty calibration.  Diversity regularization  for ensemble generation is an old idea, so is the idea of using ensemble for uncertainty calibration. The new thing here is that the authors argue that when diversity regularization is used for generating ensembles,  it helps improve the uncertainty calibration as well.
The idea can be potentially useful but the ideas are not that novel. The comparison against the baselines seem to use an overly simplified baseline without one of the critical components (the adversarial training). I am not fully convinced of the method's value based on the comparison.

---

### Official Review · AnonReviewer1 · 2018-03-12
**Unsupervised variance regularization improves calibration performance somewhat.**

**Rating:** 4
**Confidence:** 4

**Review:**

The novelty of the paper is limited: the proposed regularization, called "negative correlation" by the authors, is a flavor of variance-based regularization, well known to improve generalization (see recent work of Namkoong, Duchi, et al.) and therefore calibration, with a "proper loss" like cross-entropy.

The paper is written fairly clearly but omits some significant points. I believe a "pure ensemble" should be defined to be generated by repeatedly training the model with fixed parameter settings, but this paper never precisely defines it. This is crucial, for computation and deployment reasons, and for reproducibility.

The experimental evaluation is adequate, with its measures well-chosen to conform with the resurgence of recent work on calibration in deep learning. But I do not think the regularizer has a fresh message to the intended audience.

---

### Decision · Program_Chairs · 2018-03-20
**ICLR 2018 Workshop Acceptance Decision**

**Decision:**

Reject

**Comment:**

Based on the reviews, this paper has not been accepted for presentation at the ICLR workshop. However, the conversation and updates can continue to appear here on OpenReview.